# Boosting Trust Region Policy Optimization by normalizing flows Policy

## Abstract

We propose to improve trust region policy search with normalizing flows policy. We illustrate that when the trust region is constructed by KL divergence constraint, normalizing flows policy can generate samples far from the 'center' of the previous policy iterate, which potentially enables better exploration and helps avoid bad local optima. We show that normalizing flows policy significantly improves upon factorized Gaussian policy baseline, with both TRPO and ACKTR, especially on tasks with complex dynamics such as Humanoid.

## 1 Introduction

In on-policy optimization, vanilla policy gradient algorithms suffer from occasional updates with large step size, which lead to collecting bad samples that the policy cannot recover from (Schulman et al., 2015). Motivated to overcome such instability, Trust Region Policy Optimization (TRPO) (Schulman et al., 2015) constraints the KL divergence between consecutive policies to achieve much more stable updates. However, with factorized Gaussian policy, such KL divergence constraint can put a very stringent restriction on the new policy, making it either hard to bypass locally optimal solutions or slow down the learning process.

Can we improve the learning process of trust region policy search by using a more expressive policy class? Intuitively, a more expressive policy class has more capacity to represent complex distributions and the KL constraint may not impose very strict restriction on the sample space. Though prior works (Haarnoja et al., 2017; 2018b;a) have proposed to use implicit generative models as policies, their focus is on off-policy learning. In this work, we show how normalizing flows can be combined with on-policy learning and boost the performance of trust region policy optimization.

The structure of our paper is as follows. In Section 2 and 3, we provide backgrounds on TRPO and related work. In Section 4, we introduce normalizing flows for control and analyze why KL constraint may not impose a constraint on the sampled action space. On illustrative examples, we show that normalizing flows policy can learn policies with correlated actions and multi-modal policies, which allows for potentially more efficient exploration. In Section 5, we show by comprehensive experiment results that normalizing flows significantly outperforms baseline policy classes when combined with trust region policy search algorithms.

## 2 Background

### 2.1 Markov Decision Process

In the standard formulation of Markov Decision Process (MDP), at time step $t \geq 0$, an agent is in state $s_t \in \mathcal{S}$, takes an action $a_t \in \mathcal{A}$, receives an instant reward $r_t = r(s_t, a_t) \in \mathbb{R}$ and transitions to a next state $s_{t+1} \sim p(\cdot|s_t, a_t) \in \mathcal{S}$. Let $\pi : \mathcal{S} \mapsto P(\mathcal{A})$ be a policy, where $P(\mathcal{A})$ is a set of distribution over the action space $\mathcal{A}$. The discounted cumulative reward under policy $\pi$ is

$$J(\pi) = \mathbb{E}_\pi \Big[ \sum_{t=0}^\infty \gamma^t r_t \Big], \tag{1}$$

where $\gamma \in [0, 1)$ is a discount factor. The objective of RL is to search for a policy $\pi$ that achieves the maximum cumulative reward $\pi^* = \arg\max_\pi J(\pi)$. For convenience, we define action value function

$Q^\pi(s, a) = \mathbb{E}_\pi\big[J(\pi)|s_0 = s, a_0 = a\big]$ and value function $V^\pi(s) = \mathbb{E}_\pi\big[J(\pi)|s_0 = s, a_0 \sim \pi(\cdot|s_0)\big]$. We also define the advantage function $A^\pi(s, a) = Q^\pi(s, a) - V^\pi(s)$.

## 2.2 POLICY OPTIMIZATION

One way to search for $\pi^*$ is through direct policy search within a given policy class $\pi_\theta, \theta \in \Theta$ where $\Theta$ is the parameter space for the policy parameter. We can update the paramter $\theta$ with policy gradient ascent, by computing $\nabla_\theta J(\pi_\theta) = \mathbb{E}_{\pi_\theta}\big[\sum_{t=0}^\infty A^{\pi_\theta}(s_t, a_t)\nabla_\theta \log \pi_\theta(a_t|s_t)\big]$, then updating $\theta_{\text{new}} \leftarrow \theta + \alpha\nabla_\theta J(\pi_\theta)$ for some learning rate $\alpha > 0$. Alternatively, the update can be formulated as trust region optimization problem

$$\max_{\theta_{\text{new}}} \mathbb{E}_{\pi_\theta}\big[\frac{\pi_{\theta_{\text{new}}}(a_t|s_t)}{\pi_\theta(a_t|s_t)} A^{\pi_\theta}(s_t, a_t)\big],$$
$$||\theta_{\text{new}} - \theta||_2 \le \epsilon, \tag{2}$$

for some $\epsilon > 0$. If we do a linear approximation of the objective in (2), $\mathbb{E}_{\pi_\theta}\big[\frac{\pi_{\theta_{\text{new}}}(a_t|s_t)}{\pi_\theta(a_t|s_t)} A^{\pi_\theta}(s_t, a_t)\big] \approx \mathbb{E}_{\pi_\theta}\big[A^{\pi_\theta}(s_t, a_t)\big] + \nabla_\theta J(\pi_\theta)^T(\theta_{\text{new}} - \theta)$, we recover the gradient update by properly choosing $\epsilon$.

## 2.3 TRUST REGION POLICY OPTIMIZATION

Trust Region Policy Optimization (TRPO) (Schulman et al., 2015) applies information theoretic constraints instead of Euclidean constraints on $\theta_{\text{new}}$ and $\theta$ to better capture the geometry on the parameter space induced by the policy. In particular, consider the following trust region formulation

$$\max_{\theta_{\text{new}}} \mathbb{E}_{\pi_\theta}\big[\frac{\pi_{\theta_{\text{new}}}(a_t|s_t)}{\pi_\theta(a_t|s_t)} A^{\pi_\theta}(s_t, a_t)\big],$$
$$\mathbb{E}_s\big[\mathbb{KL}[\pi_\theta(\cdot|s)||\pi_{\theta_{\text{new}}}(\cdot|s)]\big] \le \epsilon, \tag{3}$$

where $\mathbb{E}_s\big[\,\cdot\,\big]$ is w.r.t. the state visit distribution induced by $\pi_\theta$. The trust region enforced by the KL divergence entails that the update according to (3) optimizes a lower bound of $J(\pi_\theta)$, so as to avoid accidentally taking large steps that irreversibly degrade the policy performance during training as in vanilla policy gradient (2) (Schulman et al., 2015). For a practical algorithm, the trust region constraint is approximated by a second order expansion $\mathbb{E}_s\big[\mathbb{KL}[\pi_\theta(\cdot|s)||\pi_{\theta_{\text{new}}}(\cdot|s)]\big] \approx (\theta_{\text{new}} - \theta)^T\hat{H}(\theta_{\text{new}} - \theta) \le \epsilon$ where $\hat{H} = \frac{\partial^2}{\partial\theta^2}\mathbb{E}_{\pi_\theta}\big[\mathbb{KL}[\pi_\theta(\cdot|s)||\pi_{\theta_{\text{new}}}(\cdot|s)]\big]$ is the expected Fisher information matrix. If we also linearly approximate the objective, the trust region formulation turns into a quadratic programming

$$\max_{\theta_{\text{new}}} \nabla_\theta J(\pi_\theta)^T(\theta_{\text{new}} - \theta),$$
$$(\theta_{\text{new}} - \theta)^T\hat{H}(\theta_{\text{new}} - \theta) \le \epsilon. \tag{4}$$

The optimal solution to (4) is $\propto \hat{H}^{-1}\nabla_\theta J(\pi_\theta)$. In cases where $\pi_\theta$ is parameterized by a neural network with a large number of parameters, $\hat{H}^{-1}$ is formidable to compute. Instead, (Schulman et al., 2015) proposes to approximate $\hat{H}^{-1}\nabla_\theta J(\pi_\theta)$ by conjugate gradient (CG) descent (Wright & Nocedal, 1999) since it only requires relatively cheap Hessian-vector products. Given the approximated gradient direction $\hat{g} \approx \hat{H}^{-1}\nabla_\theta J(\pi_\theta)$ obtained from CG, the KL constraint is enforced by setting $\Delta\theta = \sqrt{\frac{\epsilon}{\hat{g}^T\hat{H}\hat{g}}}\hat{g}$. Finally a line search is carried out to determine a scaler $s$ by enforcing the exact KL constraint $\mathbb{E}_{\pi_\theta}\big[\mathbb{KL}[\pi_{\theta+s\Delta\theta}||\pi_\theta]\big] \le \epsilon$ and finally $\theta_{\text{new}} \leftarrow \theta + s\Delta\theta$.

**ACKTR**   ACKTR (Wu et al.) proposes to replace the above CG descent of TRPO by Kronecker-factored approximation (Martens & Grosse, 2015) when computing the inverse of Fisher information matrix $\hat{H}^{-1}$. This approximation is more stable than CG descent and yields performance gain over conventional TRPO.

## 3 RELATED WORK

Most on-policy optimization algorithms are based on policy gradient theorem for function approximation (Sutton et al., 2000). Vanilla policy gradient algorithms are typically more stable than off-policy

learning due to their optimization based formulation, but can suffer from instability as a result of occasionally large step sizes. In policy based algorithms, updating policies with very large step sizes can be catastrophic to the learning process since the policy will collect bad samples and potentially never recover (Schulman et al., 2015). Natural policy gradient (Kakade, 2002) applies natural gradient for the policy updates, which accounts for the information geometry induced by the policy and makes the update more stable. More recently, Trust region policy optimization (Schulman et al., 2015) derives a tractable trust region policy search algorithm based on the lower bound formulation of (Kakade & Langford, 2002) and achieves promising results on simulated locomotion tasks. The trust region is approximated by the Fisher information matrix, whose inverse is further approximated by conjugate gradient iterations (Wright & Nocedal, 1999). To further improve the scalability and numerical performance of TRPO, ACKTR (Wu et al.) applies Kronecker-factored approximation (Martens & Grosse, 2015) to invert the Fisher information matrix. Orthogonal to prior works, we aim to improve TRPO with a more expressive policy representation, and we show significant improvements on both TRPO and ACKTR. We limit our attention to (Dinh et al., 2014) while other normalizing flows architectures might provide additional benefits (Kingma & Dhariwal, 2018).

A number of recent prior works have proposed to boost RL algorithms with expressive policy classes. For off-policy learning, Soft Q-learning (SQL) (Haarnoja et al., 2017) takes an implicit generative model as the policy and trains the policy by Stein variational gradient (Liu & Wang, 2016). Similarly, (Tang & Agrawal, 2018) applies an implicit policy along with a discriminator to compute entropy regularized gradient for the implicit distribution. Latent space policy (Haarnoja et al., 2018a) applies normalizing flows as the policy and displays promising results on hierarchical tasks. Soft Actor Critic (SAC) applies a mixture of Gaussian as the policy. So far, expressive policy classes have shown improvement over baselines in the domain of off-policy learning. However, it is not clear whether such benefits come from an enriched policy class or a novel algorithmic procedure. In this work, we fix the trust region search algorithms and study the net effect of expressive policy classes.

By definition, normalizing flows stacks layers of invertible transformations to map a source noise into target samples (Dinh et al., 2016; Rezende & Mohamed, 2015). Through invertible transformations, normalizing flows retains tractable probability densities while being very expressive. Normalizing flows is widely applied in probabilistic generative modeling, such as variational inference (Rezende & Mohamed, 2015). Previous works have proposed to represent policies using normalizing flows in the context of off-policy learning (Haarnoja et al., 2018a). Complement to prior works, we show that normalizing flows can significantly boost the performance of on-policy optimization.

## 4 NORMALIZING FLOWS POLICY FOR ON-POLICY OPTIMIZATION

### 4.1 NORMALIZING FLOWS FOR CONTROL

We construct a stochastic policy with normalizing flows. Normalizing flows (Rezende & Mohamed, 2015; Dinh et al., 2016) have been applied in variational inference and probabilistic modeling to represent complex distributions. In general, consider transforming a source noise $\epsilon \sim \rho_0(\cdot)$ by a series of invertible nonlinear functions $g_{\theta_i}(\cdot), 1 \leq i \leq K$ each with parameter $\theta_i$, to output a target sample $x$,

$$x = g_{\theta_K} \circ g_{\theta_{K-1}} \circ ... \circ g_{\theta_2} \circ g_{\theta_1}(\epsilon). \tag{5}$$

Let $\Sigma_i$ be the inverse of the Jacobian matrix of $g_\theta(\cdot)$, then the log density of $x$ is computed by change of variables formula,

$$\log p(x) = \log p(\epsilon) + \sum_{i=1}^{K} \log \det(\Sigma_i). \tag{6}$$

For a general invertible transformation $g_{\theta_i}(\cdot)$, computing $\det(\Sigma_i)$ is expensive. We follow the architecture of (Dinh et al., 2014) to ensure that $\det(\Sigma_i)$ is computed in linear time. To combine state information, we embed state $s$ by another neural network $L_{\theta_s}(\cdot)$ with parameter $\theta_s$ and output a state vector $L_{\theta_s}(s)$ with the same dimension as $\epsilon$. We can then insert the state vector between any two layers of (5) to make the distribution conditional on state $s$. In our implementation, we insert the state vector after the first transformation (we detail our architecture design in the Appendix B).

$$a = g_{\theta_K} \circ g_{\theta_{K-1}} \circ ... \circ g_{\theta_2} \circ (L_{\theta_s}(s) + g_{\theta_1}(\epsilon)). \tag{7}$$

Though the additive form of $L_{\theta_s}(s)$ and $g_{\theta_1}(\epsilon)$ may in theory limit the capacity of the model, in experiments below we show that the resulting policy is still very expressive. For simplicity, we denote the above transformation (7) as $a = f_\theta(s, \epsilon)$ with parameter $\theta = \{\theta_s, \theta_i, 1 \le i \le K\}$. It is obvious that the transformation $a = f_\theta(s, \epsilon)$ is still invertible between $a$ and $\epsilon$, which is critical for computing $\log \pi_\theta(a|s)$ according to (6). Such representations build complex policy distributions with explicit probability density $\pi_\theta(\cdot|s)$, and hence entail training using score function gradient estimators.

In on-policy optimizations, it is necessary to compute gradients of the entropy $\nabla_\theta \mathbb{H}[\pi_\theta(\cdot|s)]$, either for computing Hessian vector product (Schulman et al., 2015) or for entropy regularization (Schulman et al., 2015; 2017; Mnih et al., 2016). For normalizing flows there is no analytic form for entropy, we use samples to estimate entropy by re-parameterization, $\mathbb{H}[\pi_\theta(\cdot|s)] = \mathbb{E}_{a \sim \pi_\theta(\cdot|s)}[-\log \pi_\theta(a|s)] = \mathbb{E}_{\epsilon \sim \rho_0(\cdot)}[-\log \pi_\theta(f_\theta(s, \epsilon)|s)]$. The gradient of the entropy can be easily computed by a pathwise gradient and easily implemented using back-propagation $\nabla_\theta \mathbb{H}[\pi_\theta(\cdot|s)] = \mathbb{E}_{\epsilon \sim \rho_0(\cdot)}[-\nabla_\theta \log \pi_\theta(f_\theta(s, \epsilon)|s)]$.

## 4.2 Normalizing Flows Policy vs. Gaussian Policy under KL Constraint

We analyze the properties of normalizing flows policy vs. Gaussian policy under the KL constraints of trust region policy search. As a low dimensional toy example, assume we have a factorized Gaussian in $\mathbb{R}^2$ with zero mean and diagonal covariance $\mathbb{I} \cdot \sigma^2$ where $\sigma^2 = 0.1^2$. Let $\hat{\pi}_o$ be the empirical distribution formed by samples drawn from this Gaussian. We can define a KL ball centered on $\hat{\pi}_o$ as all distributions such that a KL constraint is satisfied $\mathcal{B}(\hat{\pi}_o, \epsilon) = \{\pi : \mathbb{KL}[\hat{\pi}_o||\pi] \le \epsilon\}$. We study a typical normalizing flows distribution and factorized Gaussian distribution on the boundary of such a KL ball (such that $\mathbb{KL}[\hat{\pi}_o||\pi] = \epsilon$). We find such distributions by randomly initializing the distribution parameters then running gradient updates until $\mathbb{KL}[\hat{\pi}_o||\pi]\} \approx \epsilon$. In Figure 1 (a) we show the log probability contour of such a factorized Gaussian vs. normalizing flows, and in (b) we show their samples (blue are samples from the distributions on the boundary of the KL ball and red are samples to generate $\hat{\pi}_o$). As seen from both the contour and the sample plot, though both distributions have infinite support, normalizing flows distribution has much larger variance than the factorized Gaussian, which also leads to a much larger effective support, even though both satisfy the KL constraint to the origin distribution $\mathbb{KL}[\hat{\pi}_o||\pi] = \epsilon$.

In Figure 1 (c), we show the samples drawn from factorized Gaussian and normalizing flows distribution with fixed levels of entropy $H$. To obtain distributions with fixed entropy, we randomly initialize the distribution parameters with $\mathbb{H}[\pi]$ as the entropy then obtain the desired level of entropy by minimizing $(\mathbb{H}[\pi] - H)^2$ until convergence. As seen from the sample plot (red for Gaussian and blue for normalizing flows), under the same entropy level, normalizing flows distribution has significantly larger spread than factorized Gaussian.

As analyzed above, under similar entropy level and KL constraint, normalizing flows tends to have a probability density function that decays at a much slower rate than Gaussian from the 'center', which produces a much wider effective support on the sample space. In practice, this usually produces much better exploration and helps the agent to bypass bad locally optimal solutions. For a factorized Gaussian distribution, enforcing a KL constraint does not allow the new distribution to generate samples that are too far from the 'center' of the old distribution. On the other hand, for a normalizing flows distribution, the KL constraint does not hinder the new distribution to have a very distinct support from the reference distribution (as suggested in Figure 1), hence allowing for more efficient exploration.

## 4.3 Expressiveness of Normalizing Flows Policy

We illustrate two potential strengths of the normalizing flows policy: learning correlated actions and learning multi-modal policy. First consider a 2D bandit problem where the action $a \in [-1, 1]^2$ and $r(a) = -a^T \Sigma^{-1} a$ for some positive semidefinite matrix $\Sigma$. In the context of conventional RL objective $J(\pi)$, the optimal policy is deterministic $\pi^* = [0, 0]^T$. However, in maximum entropy RL (Haarnoja et al., 2017; 2018b) where the objective is $J(\pi) + c\mathbb{H}[\pi]$, the optimal policy is $\pi_{\text{ent}}^* \propto \exp(\frac{r(a)}{c})$, a Gaussian with $\frac{\Sigma}{c}$ as the covariance matrix (red curves show the density contours). In Figure 2 (a), we show the samples generated by various trained policies to see whether they manage to learn the correlations between actions in the maximum entropy policy $\pi_{\text{ent}}^*$. We find that factorized

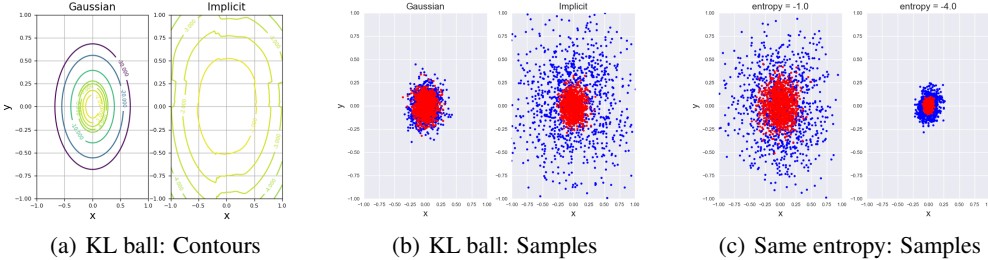

(a) KL ball: Contours          (b) KL ball: Samples         (c) Same entropy: Samples

Figure 1: Analyzing normalizing flows vs. Gaussian: (a)(b) Consider a 2D Gaussian distribution with zero mean and factorized variance $\sigma^2 = 0.1^2$. Samples from the Gaussian form an empirical distribution $\hat{\pi}_o$ (red dots in (b)) and define the KL ball $\mathcal{B}(\hat{\pi}_o, \epsilon) = \{\pi : \mathbb{KL}[\hat{\pi}_o||\pi] \leq \epsilon\}$ centered at $\hat{\pi}_o$. Find a typical normalizing flows distribution and a Gaussian distribution at the boundary of $\mathcal{B}(\hat{\pi}, 0.01)$ such that the constraint is tight. (a) Contour of log probability of a normalizing flows distribution (right) vs. Gaussian distribution (left); (b) Samples (blue dots) generated from normalizing flows distribution (right) and Gaussian distribution (left). (c) Samples generated from a normalizing flows policy (blue) vs. Gaussian policy (red) with the same entropy $H$, left panel is $H = -1.0$ and right panel $H = -4.0$.

Gaussian cannot capture the correlations due to the factorized distribution. Though Gaussian mixtures models (GMM) with $K \geq 2$ components are more expressive than factorized Gaussian, all the modes seem to collapse to the same location and suffer the same issue as factorized Gaussian. On the other hand, normalizing flows policy is much more flexible and can fairly accurately capture the correlation structure of $\pi_{\text{ent}}^*$.

To illustrate multi-modality, consider again a 2D bandit problem (Figure 2 (b)) with reward $r(a) = \max_{i \in I}\{(a - \mu_i)^T \Lambda_i^-(a - \mu_i)\}$ where $\Lambda_i, i \in I$ are diagonal matrices and $\mu_i, i \in I$ are modes of the reward landscape. In our example we set $|I| = 2$ two modes and the reward contours are plotted as red curves. Notice that GMM with varying $K$ can still easily collapse to one of the two modes while the normalizing flows policy generates samples that cover both modes.

To summarize the above two cases, since the maximum entropy objective $J(\pi) + c\mathbb{H}[\pi] = -\mathbb{KL}[\pi||\pi_{\text{ent}}^*]$, the policy search problem is equivalent to a variational inference problem where the variational distribution is $\pi$ and the target distribution is $\pi_{\text{ent}}^* \propto \exp(\frac{r(a)}{c})$. Since normalizing flows policy is a more expressive class of distribution than GMM and factorized Gaussian, we also expect the approximation to the target distribution to be much better (Rezende & Mohamed, 2015).

The properties of normalizing flows illustrated in Section 4.2 and Section 4.3 potentially allow for better exploration during training, and help bypass bad locally optimal solutions. For a more realistic example, we illustrate such benefits with the locomotion task of Ant robot (Brockman et al., 2016). In Figure 2 (c) we show the robot's 2D center-of-mass trajectories generated by normalizing flows policy (red) vs. Gaussian policy (blue) after training for $2 \cdot 10^6$ time steps. We observe that the trajectories by normalizing flows policy are much more widespread, while trajectories of Gaussian policy are quite concentrated at the initial position (the origin $[0.0, 0.0]$). Behaviorally, Gaussian policy gets the robot to jump forward quickly, which achieves high immediate rewards but terminates the episode prematurely (due to a termination condition of the task). On the other hand, normalizing flows policy bypasses such locally optimal behavior by getting the robot to move forward in a fairly slow but steady manner, even occasionally move in the opposite direction to what reward function specifies (Details in the Appendix E).

## 5 EXPERIMENTS

In experiments we aim to address the following questions: (1) Do normalizing flows policies outperform simple policies (e.g. factorized Gaussian baseline) with trust region search algorithms on benchmark tasks? (2) How sensitive are normalizing flows policies to hyper-parameters compared to Gaussian policies? To address (1), we compare normalizing flows policy against factorized Gaussian and mixture of Gaussians policies on OpenAI gym MuJoCo (Brockman et al., 2016; Todorov, 2008), rllab (Duan et al., 2016) and Roboschool Humanoid (Schulman et al., 2017) locomotion tasks as

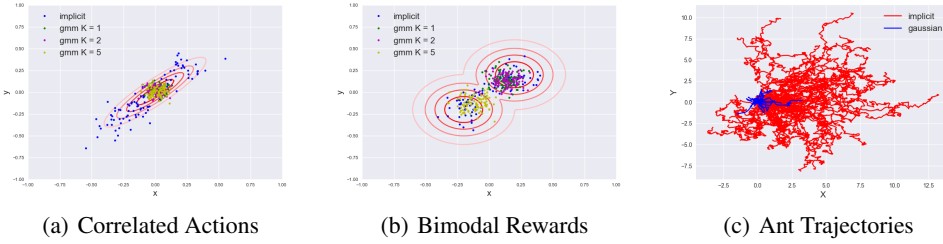

(a) Correlated Actions      (b) Bimodal Rewards      (c) Ant Trajectories

Figure 2: Expressiveness of normalizing flows policy: (a) Bandit problem with reward $r(a) = -a^T \Sigma^{-1} a$. The maximum entropy optimal policy is a Gaussian distribution with $\Sigma$ as its covariance (red contours). normalizing flows policy (blue) can capture such covariance while Gaussian cannot (green). (b) Bandit problem with multimodal reward (red contours the reward landscape). normalizing flows policy can capture the multimodality (blue) while Gaussian cannot (green). (c) Trajectories of Ant robot. The trajectories of Gaussian policy center at the initial position (the origin $[0.0, 0.0]$), while trajectories of normalizing flows policy are much more widespread.

illustrated in Figure 8. We show that normalizing flows policy can fairly uniformly outperform policies with simple distributions, especially on tasks with highly complex dynamics. We show results for both TRPO and ACKTR. To address (2), we randomly sample hyper-parameters for both normalizing flows policy and Gaussian policy, and compare their quantiles.

**Implementation Details.** As we aim to study the net effect of an expressive policy on trust region policy search, we make minimal modification to the original TRPO/ACKTR algorithms during implementations. For both algorithms, the policy entropy $\mathbb{H}\big[\pi_\theta(\cdot|s)\big]$ is analytically computed when $\pi_\theta$ is factorized Gaussian, and is estimated by samples when $\pi_\theta$ is GMM for $K \geq 2$ or normalizing flows. The KL divergence is approximated by samples instead of analytically computed in a similar way. We leave all hyper-parameter settings in the Appendix A.

### 5.1 Locomotion benchmarks

**TRPO** In Figure 3 we show the results on benchmark control problems from MuJoCo. We compare four policy classes under TRPO: factorized Gaussian (blue curves), GMM with $K = 2$ (yellow curves), GMM with $K = 5$ (green curves) and normalizing flows (red curves). For GMM, each cluster has the same probability weight and each cluster is a factorized Gaussian with independent parameters. We train each policy for a fixed number of time steps and report both mean and standard deviation of the performance averaged across 5 random seeds. We find that though GMM policies with $K \geq 2$ outperform factorized Gaussian on relatively hard tasks such as Ant and HalfCheetah, they suffer from less stable learning for Humanoid tasks. However, normalizing flows consistently outperforms GMM and factorized Gaussian policies on a wide range of tasks, especially tasks with highly complex dynamics such as Humanoid.

In Table 1, we compare with recently proposed policy classes which aim at bounding the support of policy distributions, such as Beta distribution (Chou et al., 2017) and Gaussian+tanh to bound the distribution mean. We evaluate the policies on benchmark tasks with relatively complex dynamics and show mean $\pm$ std rewards and we highlight the top two policies. We find that normalizinng flows policy consistently performs well across all complex tasks, while the performance of other policies is not as uniformly good. Also we find that bounding the final outputs by applying tanh at the final layer for normalizing flows does not perform as well, we omit the results here.

To further illustrate the strength of normalizing flows policy on Humanoid tasks, we evaluate normalizing flows vs. factorized Gaussian on Roboschool Humanoid tasks shown in Figure 4. We observe that ever since the early stage of learning (steps $\leq 10^7$) normalizing flows policy (red curves) already outperforms Gaussian (blue curves) by a large margin. In (a)(b), Gaussian is stuck in a locally optimal gait and cannot progress, while normalizing flows can bypass such locally optimal gaits and makes consistent improvement.

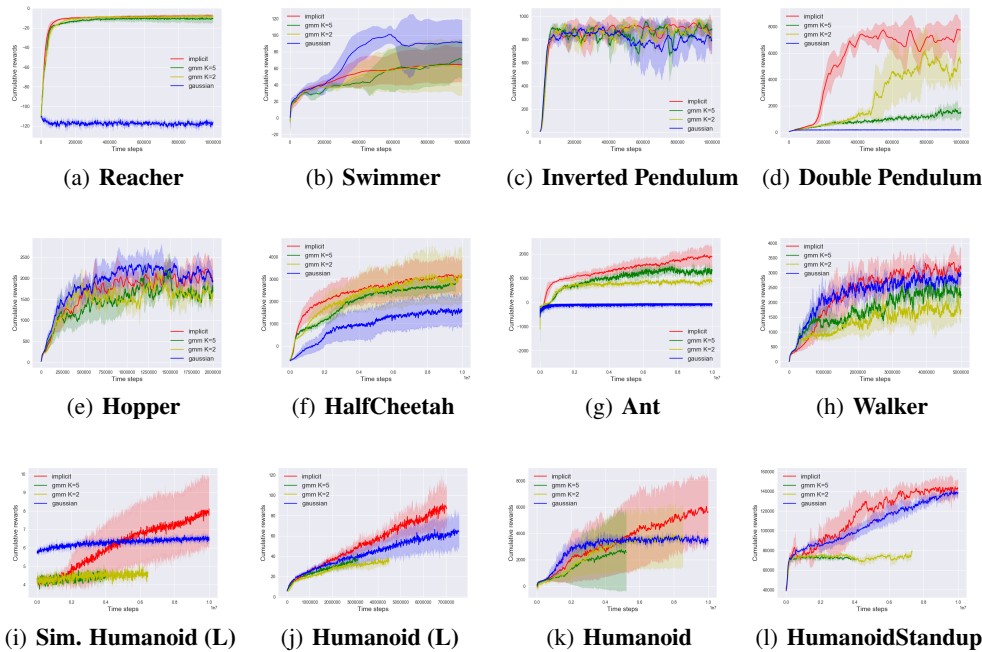

(a) **Reacher**  (b) **Swimmer**  (c) **Inverted Pendulum**  (d) **Double Pendulum**

(e) **Hopper**  (f) **HalfCheetah**  (g) **Ant**  (h) **Walker**

(i) **Sim. Humanoid (L)**  (j) **Humanoid (L)**  (k) **Humanoid**  (l) **HumanoidStandup**

Figure 3: MuJoCo Benchmark: learning curves on MuJoCo locomotion tasks. Tasks with (L) are from rllab. Each curve is averaged over 5 random seeds and shows mean ± std performance. Each curve corresponds to a different policy representation (Red: Normalizing flows (labelled as implicit), Green: GMM $K = 5$, Yellow: GMM $K = 2$, Blue: Gaussian). Vertical axis is the cumulative rewards and horizontal axis is the number of time steps.

| Tasks | Gaussian | Gaussian+tanh | Beta | NF |
|---|---|---|---|---|
| **Ant** | $-76 \pm 14$ | $-89 \pm 13$ | $\mathbf{2362 \pm 305}$ | $\mathbf{1982 \pm 407}$ |
| **HalfCheetah** | $1576 \pm 782$ | $386 \pm 78$ | $\mathbf{1643 \pm 819}$ | $\mathbf{2900 \pm 554}$ |
| **Humanoid** | $3560 \pm 288$ | $\mathbf{6350 \pm 486}$ | $3199 \pm 2222$ | $\mathbf{5222 \pm 2436}$ |
| **Humanoid (L)** | $\mathbf{64.7 \pm 7.6}$ | $38.2 \pm 2.3$ | $37.8 \pm 3.4$ | $\mathbf{87.2 \pm 19.6}$ |
| **Sim. Humanoid (L)** | $\mathbf{6.5 \pm 0.2}$ | $4.4 \pm 0.1$ | $4.2 \pm 0.2$ | $\mathbf{8.0 \pm 1.8}$ |
| **Humanoid Standup** | $\mathbf{137955 \pm 9238}$ | $133558 \pm 9238$ | $111497 \pm 15031$ | $\mathbf{142568 \pm 9296}$ |

Table 1: A comparison of various policy classes on complex benchmark tasks. For each task, we show the cumulative rewards (mean ± std) after training for $10^7$ steps across 5 seeds (for Humanoid (L) it is $7 \cdot 10^6$ steps). For each task, the top two results are highlighted in bold font.

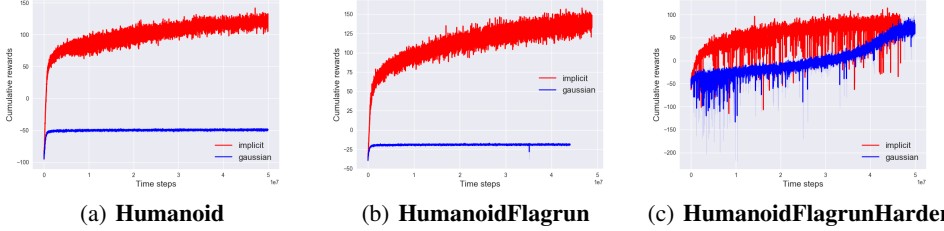

(a) **Humanoid**  (b) **HumanoidFlagrun**  (c) **HumanoidFlagrunHarder**

Figure 4: Roboschool Humanoid Benchmark : learning curves on Roboschool Humanoid locomotion tasks. Each curve corresponds to a separate random seed and we show two seeds per policy class. Each color corresponds to a different policy representation (Red: Normalizing flows (labelled as implicit), Blue: Gaussian). Vertical axis is the cumulative rewards and horizontal axis is the number of time steps.

**ACKTR**   We also evaluate different policy classes combined with ACKTR (Wu et al.). In Figure 5, we compare factorized Gaussian (red curves) against normalizing flows (blue curves) on a suite of MuJoCo and Roboschool control tasks. We train each policy for a fixed number of time steps and report both mean and standard deviation of the performance averaged across 3 random seeds. Though ACKTR + normalizing flows does not uniformly outperform Gaussian on all tasks, we find that for tasks with relatively complex dynamics (e.g. Ant and Humanoid), normalizing flows policy achieves significant performance gains. We find that the effect of an expressive policy class is fairly orthogonal to the additional training stability introduced by ACKTR over TRPO and the combined algorithm achieves even better performance.

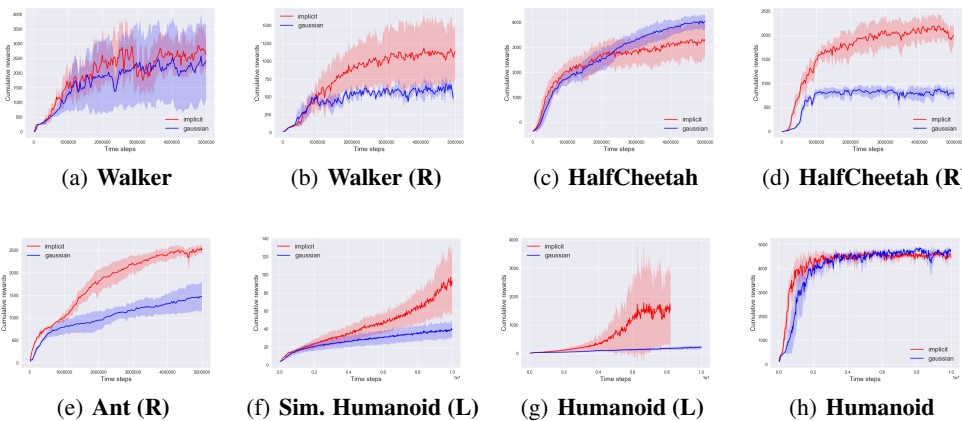

| (a) **Walker** | (b) **Walker (R)** | (c) **HalfCheetah** | (d) **HalfCheetah (R)** |
|---|---|---|---|
| (e) **Ant (R)** | (f) **Sim. Humanoid (L)** | (g) **Humanoid (L)** | (h) **Humanoid** |

Figure 5: MuJoCo and Roboschool Benchmarks : learning curves on locomotion tasks for ACKTR. Each curve is averaged over 3 random seeds and shows mean ± std performance. Each curve corresponds to a different policy representation (Red: Normalizing flows (labelled as implicit), Blue: Gaussian). Vertical axis is the cumulative rewards and horizontal axis is the number of time steps. Tasks with (R) are from Roboschool.

## 5.2   SENSITIVITY TO HYPER-PARAMETERS AND ABLATION STUDY

We evaluate the policy classes' sensitivities to hyper-parameters in Figure 6, where we compare Gaussian vs. normalizing flows. Recall that $\epsilon$ is the constant for KL constraint. For each policy, we uniformly random sample $\log_{10} \epsilon \in [-3.0, -2.0]$ and one of five random seeds, and train policies with TRPO for a fixed number of time steps. The final performance (cumulative rewards) is recorded and Figure 6 in Appendix C shows the quantile plots of final rewards across multiple tasks. We see that normalizing flows policy is generally much more robust to such hyper-parameters, importantly to $\epsilon$. We speculate that such additional robustness partially stems from the fact that for normalizing flows policy, the KL constraint does not pose very stringent restriction on the sampled action space, which allows the system to efficiently explore even when $\epsilon$ is small.

We carry out a small ablation study that addresses how hyper-parameters inherent to normalizing flows can impact the results. Recall that normalizing flows for control (Section 3) consists of $K$ transformations, with the first transformation embedding the state $s$ into a vector $L_{\theta_s}(s)$. Here we implement $L_{\theta_s}(s)$ as a two-layer neural networks with $l_1$ hidden units per layer. We evaluate on the policy performance as we vary $K \in \{2, 4, 6\}$ and $l_1 \in \{3, 5, 7\}$. We find that the performance of normalizing flows policies are fairly robust to such hyper-parameters (see Appendix C).

## 6   CONCLUSION

We propose normalizing flows as a novel on-policy architecture to boost the performance of trust region policy search. In particular, we observe that normalizing flows policy can generate samples away from the old policy while enforcing the KL constraint, which entails potentially better exploration. We evaluate performance of normalizing flows policy combined with trust region algorithms (TRPO, ACKTR) and show that they outperform factorized Gaussian and GMM policies. We propose that such policy classes be used as baselines for future benchmarking.

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

## A  HYPER-PARAMETERS

**Implementations.**  All implementations of algorithms (TRPO and ACKTR) are based on OpenAI baselines (Dhariwal et al., 2017). We implement our own GMM policy and normalizing flows policy. Environments are based on OpenAI gym (Brockman et al., 2016), rllab (Duan et al., 2016) and Roboschool (Schulman et al., 2017).

**Common Interface to the Algorithms.**  We remark that various policy classes have exactly the same interface to TRPO and ACKTR. In particular, TRPO and ACKTR only requires the computation of $\log \pi_\theta(a|s)$ (and its derivative). Different policy classes only differ in how they parameterize $\pi_\theta(a|s)$ and can be easily plugged into the algorithmic procedure originally designed for Gaussian (Dhariwal et al., 2017).

**factorized Gaussian Policy.**  A factorized Gaussian policy has the form $\pi_\theta(\cdot|s) = \mathbb{N}(\mu_\theta(s), \Sigma)$, where $\Sigma$ is a diagonal matrix with $\Sigma_{ii} = \sigma_i^2$. We use the default hyper-parameters in baselines for factorized Gaussian policy. The mean $\mu_\theta(s)$ parameterized by a two-layer neural network with 64 hidden units per layer and tanh activation function. The standard deviation $\sigma_i^2$ is each a single variable shared across all states.

**factorized Gaussian+tanh Policy.**  The architecture is the same as above but the final layer is added a tanh transformation to ensure that the mean $\mu_\theta(s) \in [-1, 1]$.

**GMM Policy.**  A GMM policy has the form $\pi_\theta(\cdot|s) = \sum_{i=1}^{K} p_i \mathbb{N}(\mu_\theta^{(i)}(s), \Sigma_i)$, where the cluster weight $p_i = \frac{1}{K}$ is fixed and $\mu_\theta^{(i)}(s), \Sigma_i$ are Gaussian parameters for the $i$th cluster. Each Gaussian has the same parameterization as the factorized Gaussian above.

**Beta Policy.**  A Beta distribution policy has the form $\pi(\alpha_\theta(s), \beta_\theta(s))$ where $\alpha_\theta(s)$ and $\beta_\theta(s)$ are shape/rate parameters parameterized by two-layer neural network $f_\theta(s)$ with a softplus at the end, i.e. $\alpha_\theta(s) = \log(\exp(f_\theta(s)) + 1) + 1$, following (Chou et al., 2017). Actions sampled from this distribution have a strictly finite support. We notice that this parameterization introduces potential instability during optimization: for example, when we want to converge on policies that sample actions at the boundary, we require $\alpha_\theta(s) \to \infty$ or $\beta_\theta(s) \to \infty$, which might be very unstable. We also observe such instability in practice.

**normalizing flows Policy.**  A normalizing flows policy has a generative form: the sample $a \sim \pi_\theta(\cdot|s)$ can be generated via $a = f_\theta(s, \epsilon)$ with $\epsilon \sim \rho_0(\cdot)$. The detailed architecture of $f_\theta(s, \epsilon)$ is in the appendix below.

**Others.**  Value functions are all parameterized as two-layer neural networks with 64 hidden units per layer and tanh activation function. Trust region sizes are enforced via a constraint parameter $\epsilon$, where $\epsilon \in \{0.01, 0.001\}$ for TRPO and $\epsilon \in \{0.02, 0.002\}$ for ACKTR. All other hyper-parameters are default parameters from the baselines implementations.

## B  NORMALIZING FLOWS POLICY ARCHITECTURE

We design the neural network architecture following the idea of (Dinh et al., 2014; 2016). Recall that normalizing flows (Rezende & Mohamed, 2015) consists of layers of transformation as follows ,

$$x = g_{\theta_K} \circ g_{\theta_{K-1}} \circ ... \circ g_{\theta_2} \circ g_{\theta_1}(\epsilon),$$

where each $g_{\theta_i}(\cdot)$ is an invertible transformation. We focus on how to design each atomic transformation $g_{\theta_i}(\cdot)$. We overload the notations and let $x, y$ be the input/output of a generic layer $g_\theta(\cdot)$,

$$y = g_\theta(x).$$

We design a generic transformation $g_\theta(\cdot)$ as follows. Let $x_I$ be the components of $x$ corresponding to subset indices $I \subset \{1, 2...m\}$. Then we propose as in (Dinh et al., 2016),

$$y_{1:d} = x_{1:d}$$
$$y_{d+1:m} = x_{d+1:m} \odot \exp(s(x_{1:d})) + t(x_{1:d}), \tag{8}$$

where $t(\cdot), s(\cdot)$ are two arbitrary functions $t, s : \mathbb{R}^d \mapsto \mathbb{R}^{m-d}$. It can be shown that such transformation entails a simple Jacobien matrix $|\frac{\partial y}{\partial x^T}| = \exp(\sum_{j=1}^{m-d}[s(x_{1:d})]_j)$ where $[s(x_{1:d})]_j$ refers to the $j$th component of $s(x_{1:d})$ for $1 \leq j \leq m - d$. For each layer, we can permute the input $x$ before apply the simple transformation (8) so as to couple different components across layers. Such coupling entails a complex transformation when we stack multiple layers of (8). To define a policy, we need to incorporate state information. We propose to preprocess the state $s \in \mathbb{R}^n$ by a neural network $L_{\theta_s}(\cdot)$ with parameter $\theta_s$, to get a state vector $L_{\theta_s}(s) \in \mathbb{R}^m$. Then combine the state vector into (8) as follows,

$$z_{1:d} = x_{1:d}$$
$$z_{d+1:m} = x_{d+1:m} \odot \exp(s(x_{1:d})) + t(x_{1:d})$$
$$y = z + L_{\theta_s}(s). \tag{9}$$

It is obvious that $x \leftrightarrow y$ is still bijective regardless of the form of $L_{\theta_s}(\cdot)$ and the Jacobien matrix is easy to compute accordingly.

In locomotion benchmark experiments, we implement $s, t$ both as 4-layers neural networks with $l_1 = 3$ units per hidden layer. We stack $K = 4$ transformations: we implement (9) to inject state information only after the first transformation, and the rest is conventional coupling as in (8). $L_{\theta_s}(s)$ is implemented as a feedforward neural network with 2 hidden layers each with 64 hidden units. Value function critic is implemented as a feedforward neural network with 2 hidden layers each with 64 hidden units with rectified-linear between hidden layers.

## C  SENSITIVITY TO HYPER-PARAMETERS AND ABLATION STUDY

In Figure 7, we show the ablation study of normalizing flows policy. We evaluate how the training curves change as we change the hyper-parameters of normalizing flows: number of transformation $K \in \{2, 4, 6\}$ and number of hidden units $l_1 \in \{3, 5, 7\}$ in the embedding function $L_{\theta_s}(\cdot)$. We find that the performance of normalizing flows policy is fairly robust to changes in $K$ and $l_1$. When $K$ varies, $l_1$ is set to 3 by default. When $l_1$ varies, $K$ is set to 4 by default.

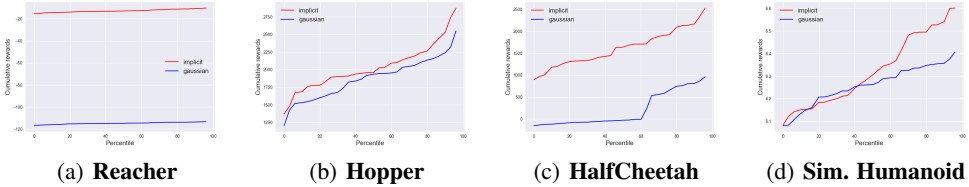

(a) **Reacher**      (b) **Hopper**      (c) **HalfCheetah**      (d) **Sim. Humanoid**

Figure 6: Sensitivity to Hyper-parameters: quantile plots of policies' performance on MuJoCo benchmark tasks under various hyper-parameter settings. For each plot, we randomly generate 30 hyper-parameters for the policy and train for a fixed number of time steps. Reacher for $10^6$ steps, Hopper and HalfCheetah for $2 \cdot 10^6$ steps and SimpleHumanoid for $\approx 5 \cdot 10^6$ steps. normalizing flows policy is in general more robust to Gaussian policy.

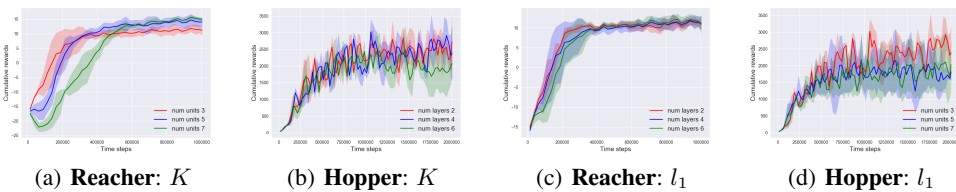

(a) **Reacher**: $K$      (b) **Hopper**: $K$      (c) **Reacher**: $l_1$      (d) **Hopper**: $l_1$

Figure 7: Sensitivity to normalizing flows Hyper-parameters: training curves of normalizing flows policy under different hyper-parameter settings (number of hidden units $l_1$ and number of transformation $K$, on Reacher and Hopper task. Each training curve is averaged over 5 random seeds and we show the mean $\pm$ std performance. Vertical axis is the cumulative rewards and horizontal axis is the number of time steps.)

## D    ILLUSTRATION OF LOCOMOTION TASKS

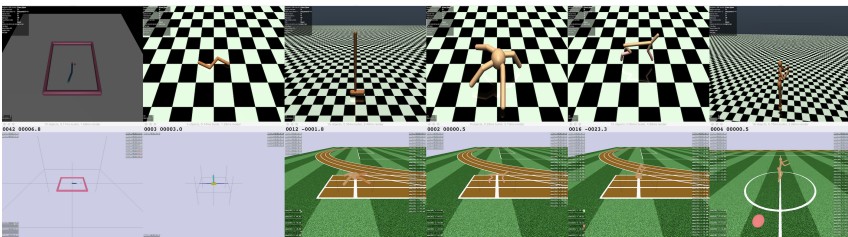

Figure 8: Illustration of benchmark tasks in OpenAI MuJoCo (Brockman et al., 2016; Todorov, 2008), rllab (top line) (Duan et al., 2016) and Roboschool (bottom line) (Schulman et al., 2017).

## E    REWARD STRUCTURE OF ANT LOCOMOTION TASK

For Ant locomotion task (Brockman et al., 2016), the state space $\mathcal{S} \subset \mathbb{R}^{116}$ and action space $\mathcal{A} \subset \mathbb{R}^8$. The state space consists of all the joint positions and joint velocities of the Ant robot, while the action space consists of the torques applied to joints. The reward function at time $t$ is $r_t \propto v_x$ where $v_x$ is the center-of-mass velocity of along the x-axis. In practice the reward function also includes terms that discourage large torques and encourages the Ant robot to stay alive (as defined by not flipping itself over).

Intuitively, the reward function encourages the robot to move along x-axis as fast as possible. This is reflected in Figure 2 (c) as the trajectories (red) generated by the normalizing flows policy is spreading along the x-axis. Occasionally, the robot also moves in the opposite direction.

