# OpenReview forum: "Boosting Trust Region Policy Optimization by Normalizing flows Policy"
_ICLR.cc/2019/Conference_

### Official Review · AnonReviewer2 · 2018-11-01
**Combining normalizing flows and Gaussian policies is relatively new, but justification is very limited**

**Rating:** 4
**Confidence:** 4

**Review:**

The papers proposed to use normalizing flow policies instead of Gaussian policies to improve exploration and achieve better sample complexity in practice. While I believe this idea has not specifically been tried in previous literature and the vague intuition that NF leads to more exploration that helps learning a better policy, the novelty of combining these two seems limited, and the paper does not seem to provide enough justification to using NF policies instead of alternative policy distributions both in theory and in the experiments.

1. About Section 4.2. I believe that the normalizing flow in question would transform the volume of a Gaussian? So there would exist some parameter setting for a flow model that also shrinks volume, thereby resulting in lower variance policies? The arguments would thereby depend heavily on the specific architecture and initialization of the flow model, which is not discussed in detail.

Also, why is finding a high variance policy better in terms of the trust region argument? Isn't the whole point of using trust region that the new policy should be closer to old policy to prevent performance degradation? I also think that a fair comparison would be compare KL between normalizing flow policies, instead of KL between NF and Gaussian.

2. The TRPO experiments seem wrong -- at least the results don't match what is reported in the ACKTR paper for Reacher and InverseDoublePendulum envs -- there the TRPO policy at least learns something. Also TRPO in general does not perform as bad as it may seem, see "Deep RL that matters" paper by Henderson et al. Maybe this is because of using OpenAI baselines code which seems to have worse TRPO performance.

There is also no experiments on ACKTR on the small Mujoco tasks (even in the Appendix), which seems to be a rather big oversight given the authors have already done even harder tasks for ACKTR + NF.

Moreover I think a fair comparison is to use almost the same architecture for implicit and gaussian, where the only difference is where you sample the noise. For Gaussian with flows, you can first use an MLP to produce deterministic outputs and then use flow to generate the mean actions. Otherwise it is impossible to say whether the architecture or the implicit distribution contributes more to the success.

One could also use truncated Gaussian distributions / Beta distributions / Gaussian + tanh, since Mujoco actions beyond (-1, 1) is treated as -1 or 1, so Gaussian should already be bad. It is unclear whether NF is able to outperform these settings.

Minor points:

- Fix citations. Please use \citep throughout.
- Is Equation (6) correct? Seems like \Sigma_i should be the inverse of g_i(\epsilon)? Also this is the "change of variables formula" not "chain rule".
- Why is normalizing flow not part of the background?
- Add legends in Figure (1)
- Figure 2(c), I believe with max entropy you could already obtain diverse ant trajectories?
- I believe in the context of generative models, "implicit" typically means the case where likelihood is not tractable? Here the likelihood is perfectly tractable.

---

> ### Author Response · Authors · 2018-11-11
> **Thank you for your review!**
>
> Thank you for your time and efforts in reviewing our paper! We have made edits to the original manuscript based on your valuable feedbacks, and would like to make a few clarifications.
>
> >>About Section 4.2. I believe that the normalizing flow in question would transform the volume of a Gaussian? So there would exist some parameter setting for a flow model that also shrinks volume, thereby resulting in lower variance policies? The arguments would thereby depend heavily on the specific architecture and initialization of the flow model, which is not discussed in detail.
>
> Thank you for pointing this out! We agree that the properties of the NF model depends on the architecture, and that using other flows architecture might result in more extensive benefits (expressive policy plus lower variance gradient). In this work we limit our attention to the architecture proposed in Dinh et al (2016) and we have made it clear in both Section 3 and 4 that we limit the attention to Dinh et al (2016) while other flows architecture might provide additional benefits.
>
> >>Also, why is finding a high variance policy better in terms of the trust region argument? Isn't the whole point of using trust region that the new policy should be closer to old policy to prevent performance degradation? I also think that a fair comparison would be compare KL between normalizing flow policies, instead of KL between NF and Gaussian.
>
> Schulman et al (2015) shows the benefit of explicitly imposing KL divergence constraint on consecutive updates, hence proposing TRPO. In our paper, we argue that while NF and Gaussian can both enforce the same constraints on KL divergence, the induced constraints on the sampled action space is different. This is critical since though the constraint is placed on KL divergence, it is the sampled actions that lead to exploration. We argue that under the same KL constraint, NF can sample actions that potentially lead to better exploration, this is also the motivation behind the comparison of sampled actions between Gaussian and NF under the same KL constraint.
>
> >>The TRPO experiments seem wrong -- at least the results don't match what is reported in the ACKTR paper for Reacher and InverseDoublePendulum envs -- there the TRPO policy at least learns something. Also TRPO in general does not perform as bad as it may seem, see "Deep RL that matters" paper by Henderson et al. Maybe this is because of using OpenAI baselines code which seems to have worse TRPO performance.
>
> Thank you for pointing this out. We are aware that deep RL experiments depend heavily on implementations. For fair comparison, throughout the paper we use OpenAI baseline implementation: all hyper-parameters are shared across various policy classes, except for their individual parameterization of pi_\theta(a|s). It is possible that baseline implementation gives poor performance on reacher and inverted-pendulum, however, due to the uniform implementation we think our comparison is fair.
>
> >>There is also no experiments on ACKTR on the small Mujoco tasks (even in the Appendix), which seems to be a rather big oversight given the authors have already done even harder tasks for ACKTR + NF.
>
> We have run similar experiments on smaller tasks but find that ACKTR + NF does not yield significant gains compared to on complex tasks. This is also consistent with our argument that complex distributions lead to performance gains on tasks with complex dynamics.
>
> >>One could also use truncated Gaussian distributions / Beta distributions / Gaussian + tanh, since Mujoco actions beyond (-1, 1) is treated as -1 or 1, so Gaussian should already be bad. It is unclear whether NF is able to outperform these settings.
>
> We have included new experiments on Beta distribution and Gaussian distribution + tanh. The motivations for these distributions are different from ours, in that they aim to bound the actions to the rang [-1,1]. We have added their results for clarity. We have also experimented with NF + tanh (at the last layer) to bound actions in [-1,1] but found it not to be helpful.
>
> >>Minor points
>
> We have fixed citation and symbol issues addressed by the reviewer.
>
> >>Why is normalizing flow not part of the background?
>
> We have introduced the generative process of NF in Section 4, which we think can serve as part of the background.
>
> >>Figure 2(c), I believe with max entropy you could already obtain diverse ant trajectories?
>
> We try to argue that NF can do better exploration than Gaussian for this task, as a result the trajectories of NF are more diverse. It is possible that adding entropy to Gaussian can lead to slightly more entropic trajectories, but we suspect that it significantly helps to escape the locally optimal gait.

---

### Official Review · AnonReviewer1 · 2018-11-02
**simple idea but not totally clear**

**Rating:** 4
**Confidence:** 4

**Review:**

This paper generalizes basic policy gradient methods by replacing the original Gaussian or Gaussian mixture policy with a normalizing flow policy, which is defined by a sequence of invertible transformations from a base policy.

Although the concept of normalizing flow is simple, and it has been applied to other models such as VAE, there seems no work on applying it for policy optimization. Thus I think this method is itself interesting.

However, I find the paper written in a way assuming readers very familiar with related concept and algorithms in reinforcement learning. Thus although one can get the general idea on how the method works, it might be difficult to get a deeper understanding on some details.

For example, normalizing flows are defined in Section 4, and then it is directly claimed that normalizing flows can be applied to policy optimization, without giving details on how it is actually applied, e.g., what is the objective function? and why one needs to compute gradients of the entropy (Section 4.1)?

Also, in the experiments, it is said that one can combing normalizing flows with TRPO without describing the details. I can't get how exactly normalizing flows + TRPO works.

The experiments also talk about 2D bandit problem, and again, without any descriptions. BTW, in the Section 4.3, what does [-1, 1]^2 mean? (I have seen {-1, 1}^2, but not [-1, 1]^2).

It seems that the authors only use the basic normalizing flow structures studied in Rezende&Mohamed (2015) and Dinh et al (2016). However, there are more powerful variants of normalizing flows such as the Multiplicative Normalizing Flows or the Glow. I wonder how good the results are if these more advanced versions are used. Maybe they can uniformly outperform Gaussian policy?

Update:
I feel the idea of this paper is straightforward, and the contribution is incremental. To improve the paper, stronger experiments need to be performed.

---

> ### Author Response · Authors · 2018-11-11
> **Thank you for your review!**
>
> Thank you for your time and efforts in reviewing our paper! We have made edits to the original manuscript based on your valuable feedbacks, and would like to make a few clarifications.
>
> >>However, I find the paper written in a way assuming readers very familiar with related concept and algorithms in reinforcement learning. Thus although one can get the general idea on how the method works, it might be difficult to get a deeper understanding on some details.
>
> We think we have made the presentation relatively clear. In terms of the algorithmic procedure, we have concisely yet comprehensively illustrated how TRPO works; in terms of architectures, we have briefly explained how NF works and detail architectures in the Appendix. We also think replacing Gaussian by NF in the context of TRPO is fairly straightforward as explained below.
>
> >>For example, normalizing flows are defined in Section 4, and then it is directly claimed that normalizing flows can be applied to policy optimization, without giving details on how it is actually applied, e.g., what is the objective function?
>
> Indeed we have assumed that transferring the policy from Gaussian to Normalizing flows (NF) is quite straightforward, in that both assume a distribution \pi_\theta(a|s) parameterized by some policy \theta. NF and Gaussian only differ in this parameterization and their interface with the algorithms is exactly the same. The objective is still the old surrogate loss of the policy optimization procedure.
>
> >>and why one needs to compute gradients of the entropy (Section 4.1)?
>
> We have mentioned that sometimes entropy regularization is needed, and for that we need to be able to compute gradient of the policy entropy.
>
> >>Also, in the experiments, it is said that one can combing normalizing flows with TRPO without describing the details. I can't get how exactly normalizing flows + TRPO works.
>
> As mentioned above, we have assumed that replacing Gaussian by NF in the case of TRPO is quite straightforward: NF and Gaussian have different parameterizations for \pi_\theta(a|s), but other than that everything else if pretty much the same. We have added a concise explanation in Appendix A.
>
> >>The experiments also talk about 2D bandit problem, and again, without any descriptions.
>
> We have described the setup for the 2D bandit problem in Section 4.3. In particular, we interpret 2D bandit problem as a state-less one-step MDP where we can more clearly compare the properties of various policy classes.
>
> >>BTW, in the Section 4.3, what does [-1, 1]^2 mean?
>
> By [-1,1]^2 we mean a 2-dimensional box: the Cartesian product of two [-1,1] intervals.
>
> >>It seems that the authors only use the basic normalizing flow structures studied in Rezende&Mohamed (2015) and Dinh et al (2016). However, there are more powerful variants of normalizing flows such as the Multiplicative Normalizing Flows or the Glow. I wonder how good the results are if these more advanced versions are used. Maybe they can uniformly outperform Gaussian policy?
>
> Thank you for pointing this out. These more advanced flows architectures are definitely worth experimenting with. In this work we only consider the architecture adopted in Dinh et al (2016) and hope to illustrate how an expressive policy improves upon trust region policy search baselines. We also think that NF policies enjoy more advantages on tasks with complex dynamics (e.g. to better escape locally optimal solutions). For tasks with simple dynamics, Gaussian might be already good enough and a complex distribution will even take longer to converge.
> In addition, we have added in Section 3 and 4 that we limit our attention to the architectures in Dnih et al (2016) while more recent advances in NF can provide more benefits.

---

### Official Review · AnonReviewer3 · 2018-11-03
**Simple modification with relatively robust gains**

**Rating:** 6
**Confidence:** 4

**Review:**

The authors in this work present an approach to policy optimization that relies on an alternative policy formulation based on normalizing flows. This is a relatively simple modification (this is no criticism) that essentially uses the same TRPO algorithm as previous approaches, but a different mechanism for generating the distribution over actions. The crux of the authors’ approach is detailed in equations (6) and (7), although it could have been useful to see more of the discussion of the architecture from appendix B in the actual text of the paper.

The authors then go on to analyze the properties and expressiveness of the resulting properties and show that it is more capable of capturing complex interactions than a simple Gaussian. It was somewhat unclear, however, in section 4.2 what the exact form of the policies being compared are. Is this a simple example with only the parameters of the Gaussian, or was the Gaussian parameterized by a multi-layer model? Further, one thing I would also have liked to see the authors question more is, for the problems they attack, whether this expressiveness is more useful “during exploration” or for the ultimate performance of the final policy.

The authors, finally, show that this approach is able to out-perform the alternative Gaussian policy. Ultimately this approach seems to be a simple modification (or replacement) of the standard policy formulation, and one that seems to lead to good performance gains.

---

> ### Author Response · Authors · 2018-11-12
> **Thank you for your review!**
>
> Thank you for your time and efforts in reviewing our paper! We have made edits to the original manuscript based on your valuable feedbacks, and would like to make a few clarifications.
>
> >>The crux of the authors’ approach is detailed in equations (6) and (7), although it could have been useful to see more of the discussion of the architecture from appendix B in the actual text of the paper.
>
> Thank you for pointing this out! As also mentioned by the other two reviewers, we agree that the properties of the NF policy depends on the architectures. In this our we focus our attention on the architectures in Dinh et al (2016) and aim to illustrate how the expressive policy derived from such specific architecture benefits trust region policy search. To better clarify the scope, we have added in Section 3 and 4 that we limit our attention to Dinh et al (2016) architecture only and other recent advances might benefit the policy search in other ways.
>
> >>The authors then go on to analyze the properties and expressiveness of the resulting properties and show that it is more capable of capturing complex interactions than a simple Gaussian. It was somewhat unclear, however, in section 4.2 what the exact form of the policies being compared are. Is this a simple example with only the parameters of the Gaussian, or was the Gaussian parameterized by a multi-layer model?
>
> It is a Gaussian with mean/std parameterized by multi-layer network. We will clarify this in our final manuscript. We thank the reviewer for pointing out studying the actual effect of the policy class on exploration. We actually tried to illustrate this issue a bit with the ant example in Section 4, where we can infer that NF explores better by observing its diverse trajectories. On the other hand, we see that Gaussian is stuck (both in rewards and trajectories), hence we infer that Gaussian explores less efficiently. In general, we think it is quite hard to disentangle the effect of exploration from the final performance of the policy. This is definitely one way to further explore how expressive policy interacts with the learning procedure, and we will pursue this as future work.

---

### Meta-Review · Area_Chair1 · 2018-12-14
**Marginal contribution, need stronger experiments**

**Confidence:** 5
**Recommendation:** Reject

**Metareview:**

This work proposes to improve trust region policy search (TRPO) by using normalizing flow policies. This idea is a straightforward combination of two existing techniques and is not super surprising in terms of novelty. In this case, really strong experiments are needed to support the work; this is , unfortunately, is the not the case.  For example, it was notice by the reviewers that the Mujoco TRPO experiments does not use the best implementation of TRPO, which makes it difficult to judge the strength of the work compared with state of the art.